# Explosive Strength Modeling in Children: Trends According to Growth and Prediction Equation

**Vittoria Carnevale Pellino** [1,2,†] , **Matteo Giuriato** [3,†] , **Gabriele Ceccarelli** [4,*] , **Roberto Codella** [5,6] , **Matteo Vandoni** [1] , **Nicola Lovecchio** [1,5] and **Alan M. Nevill** [7]

1. Laboratory of Adapted Motor Activity (LAMA), Department of Public Health, Experimental Medicine & Forensic Science, University of Pavia, 27100 Pavia, Italy; vittoria.carnevalepellino@unipv.it (V.C.P.); matteo.vandoni@unipv.it (M.V.); nicola.lovecchio@unipv.it (N.L.)
2. Department of Industrial Engineering, University of TorVergata, 00133 Rome, Italy
3. Department of Neurosciences, Biomedicine and Movement Sciences, Università di Verona, 37129 Verona, Italy; matteo.giuriato@univr.it
4. Human Anatomy Institute, Department of Public Health, Experimental Medicine & Forensic Science, University of Pavia, 27100 Pavia, Italy
5. Department of Biomedical Science for Health, Università degli Studi di Milano, 20133 Milano, Italy; roberto.codella@unimi.it
6. Department of Endocrinology, Nutrition and Metabolic Diseases, IRCCS MultiMedica, 20138 Milano, Italy
7. Faculty of Education, Health and Wellbeing, University of Wolverhampton, Walsall WV1 1SB, UK; a.m.nevill@wlv.ac.uk
* Correspondence: gabriele.ceccarelli@unipv.it; Tel.: +39-0382987661
† These authors contribute equally to this work.

**Abstract:** Lower limb explosive strength has been widely used to evaluate physical fitness and general health in children. A plethora of studies have scoped the practicality of the standing broad jump (SBJ), though without accounting for body dimensions, which are tremendously affected by growth. This study aimed at modeling SBJ-specific allometric equations, underlying an objectively predictive approach while controlling for maturity offset (MO). A total of 7317 children (8–11 years) were tested for their SBJs; demographics and anthropometrics data were also collected. The multiplicative model with allometric body size components, MO, and categorial differences were implemented with SBJ performance. The log-multiplicative model suggested that the optimal body shape associated with SBJs is ectomorphic (H = −0.435; M = 1.152). Likewise, age, sex, and age–sex interactions were revealed to be significant ($p < 0.001$). Our results confirmed the efficacy of the allometric approach to identify the most appropriate body size and shape in children. Males, as they mature, did not significantly augment their performances, whereas females did, outperforming their peers. The model successfully fit the equation for SBJ performance, adjusted for age, sex, and MO. Predictive equations modeled on developmental factors are needed to interpret appropriately the performances that are used to evaluate physical fitness.

**Keywords:** allometry; standing broad jump; children; growth; maturity offset

## 1. Introduction

The relevance of muscular strength and power is well recognized in human performance [1,2] and contributes to bone health across different age groups [3]. Several reviews highlighted health outcomes associated with muscular strength and showed that poor muscular strength in children and adolescents is associated with cardiovascular disease, metabolic profiles, skeletal health, and adiposity [4,5]. Other studies have shown that high muscular strength in youths is associated with better lipid

metabolic profiles, independent of cardiorespiratory fitness [3]. Ideally, muscular power (or explosive strength, as it is known in practical context) should be measured in lab settings [6], yielding valid values of muscular outcomes. Nevertheless, sometimes this may lack feasibility, and therefore field tests like the standing broad jump (SBJ) are preferable, since they are recognized as acceptable alternatives to lab assessments [7]. In addition, musculoskeletal fitness evaluation may enrich individual feedback to students and athletes, providing comparisons to international references on performance and health. For this reason, several youth fitness batteries [7–9] incorporated SBJ testing into their evaluation panel. Normative values for children and adolescents were created from these validation studies, proposing 50th percentiles as a proper medium level of performance that one could refer to. However, this procedure seems to have some limitations. For example, in Tomkinson et al. [10], the SBJ showed a mean improvement of 9 cm between 9–10 and 10–11 years (both males and females), while no information was obtained about changes in body mass and height, which differ according to age group and sex.

This stratification is useful for primarily surveillance purposes [11], but not for accurately monitoring training or developmental aspects. Moreover, childhood is a crucial period for detecting sensitive increases in performance (+10–15% per year) [12,13]. In children 7–8 years of age, SBJ performance is strictly dependent on body mass and height [14]. Although Body Mass Index (BMI) is the parameter summarizing the two variables, it does not represent a proper predictor of authentic performance [15,16]. Indeed, growth does not reflect parallel increases in this developmental pattern; while height increases by 3–4% per year, body mass increases by 8–10% [17]. Furthermore, a commonly used ratio scaling—the power-to-body mass ratio—may be a misleading index [18]. An alternative could be represented by the stratification of performance according to social condition [19] or BMI categories [16]. However, as mentioned earlier, this could result in a partial analysis. To convey these conceptual bottlenecks to a convincing ground of discussion, it is necessary to scale for maturity differences and determine the role of body size on performance [20], which is the allometric model. The objective of scaling is to normalize the performance for anthropometric characteristics [21]. In this framework, it is crucial to consider an adjunct polynomial model to analyze data and appropriately interpret them, discerning the contribution of developmental growth and maturation, among other factors, considering the exponential trend of human growth [22,23]. In the performance evaluation, another critical assessment item during the child growing phase could be the maturity offset (MO) at the peak of maturity. To this end, the peak high velocity (PHV) is capable of capturing a peculiar significance of speed and power results throughout growth, as well as the MO in strength (explosive strength) [16] or endurance outcomes [15]. In particular, the strength outcome depends on the developmental stage of the physiological determinants, whereas the peak of strength corresponds to the PHV [24]. Indeed, absolute values of specific strength actions, such as jumping or changing directions, are differently positioned in peak performance velocity curves as to PHV [25]. As such, controlling maturity is of paramount importance [26].

This study aims to model strength test-specific allometric equations, considering multiple individual factors such as height, body mass, sex, and age. Predictive equations will enforce a robust reliable indication of fitness per the structure and maturation levels of children (8–11 years old).

## 2. Materials and Methods

### 2.1. Experimental Approach to the Problem

During childhood, manifold factors could ameliorate performance, especially age and sex, according to the MO. In particular, the strength outcome depends on the developmental stage of the physiological determinants, whereas the peak of strength corresponds to PHV, where absolute values of specific strength actions are differently positioned in the peak performance velocity curves as compared with PHV. In this study, we decided to investigate the muscular power related to actual children's maturity, taking into account multiple individual proxies such as height, body mass, sex,

and age. Children performed an SBJ as per the field test guidelines. The biological maturation (MO) was estimated using the somatic maturation method proposed by Werneck et al. [27]. Then, the age of PHV was determined by subtracting the MO from the chronological age. Participants were classified as late, early, or on time through the one-SD method derived from the current sample.

## 2.2. Subjects

This cross-sectional study involved 7317 children (3627 girls) aged 8–11 years, recruited from 119 Northeast Italian primary schools (third, fourth, and fifth grade). Children with known chronic cardiac, respiratory, neurological, or musculoskeletal disorders were excluded. All the described measures were taken during physical education classes as scheduled in the morning activities (8:00–12:00 a.m.). The study protocol, including each feature of the experimental design, was approved by the ethical boards of the enrolled schools in accordance with the World Medical Association Declaration of Helsinki, as revised in 1983. All participants were free to withdraw their participation at any time. Written informed consent was obtained from the parents or legal guardians, while verbal assent was obtained from the children prior to participation.

## 2.3. Procedures

Data collection included demographic and anthropometric information (sex, age, mass, and height) measured before the test sessions using standardized techniques (Table 1). Height was measured using a portable stadiometer with a precision of ± 1 mm (Stadiometer Seca 213, Intermed S.r.l. Milan, Italy) with children in an upright position, with bare feet placed slightly apart, arms extended, and head positioned in the Frankfort plane. Body mass was assessed using a beam scale with a precision of ±100 g (Seca 813, Intermed S.r.l. Milan, Italy), with children in light clothing, without shoes, and stood upright at the center of the platform of the mass scale. We calculated the age of the children from the birth date and subsequently rounded down values.

**Table 1.** Demographic and anthropometric information by sex. Numerosity following age: 8–9 y (1175 males, 1074 females), 9–10 y (1240 males, 1200 females), and 10–11 y (1275 males, 1353 females). All values are showed as mean ± DS.

| Subjects | Numerosity | Age (y) | Height (cm) | Mass (kg) |
|----------|-----------|---------|-------------|-----------|
| Total | 7317 | 9.4 ± 1 | 136.5 ± 8.5 | 33.8 ± 8.4 |
| Males | 3690 | 9.4 ± 1 | 136.4 ± 8.1 | 34.0 ± 8.2 |
| Females | 3627 | 9.4 ± 1 | 136.6 ± 8.9 | 33.7 ± 8.5 |

The explosive strength of the children was measured by the SBJ test (systematic error nearly to 0) [5], a practical, time-efficient, and low-cost field test widely adopted in school or gym contexts [7,11,16,28]. Every child jumped for distance from a standstill. During the performance of the jumps, the children were asked to bend their knees with their arms in front of them, parallel to the ground, then swing both arms, push off vigorously, and jump as far as possible. The test was performed three times, scored in centimeters, and the best value was recorded. The score was obtained by measuring the distance between the last heel mark and the take off line [29].

All tests were conducted by a team of three students of the sport science degree course during scheduled physical education classes in the morning. Previous training and calibration of the operators were performed to ensure the accuracy and repeatability of the procedure. The presence and collaboration of the curricular teachers were guaranteed at any time to meet the confidence of the students [30].

## 2.4. Statistical Analysis

The biological maturation (MO) was estimated through the somatic maturation method proposed by Werneck et al. [27].

This method estimates the MO from stature and chronological age using an algorithm, providing the result offset in years to peak height velocity (PHV). In particular, this easy approach was adopted because, briefly, involving only one anthropometric measure reduces operator-dependent errors, especially because it meets the full compliance of children and parents and avoids other maturity scales (i.e., Tanner; Marshall & Tanner 1969, 1970). In brief, the MO was determined using a specific formula for girls and boys:

Maturity offset for girls (years) = −7.709133 + (0.0042232 × (age × height))

Maturity offset for boys (years) = −7.999994 + (0.0036124 × (age × height))

Then, the age of PHV was determined by subtracting the maturity offset from the chronological age. Participants were classified as late, early, or on time through the one-SD method derived from the current sample.

The multiplicative model with allometric body size components of Body Mass (M) and Height (H) was used to identify the most appropriate body size and shape characteristics (Equation (1)) associated with, as well as detect any maturity (using MO) and categorical differences (e.g., sex, age) in, the physical performance variable (SBJ). The model is an extension of the one used to predict the physical performance variables of Greek children [23]:

$$Y = a \cdot Mk^1 \cdot Hk^2 \cdot \exp(c \cdot MO) \cdot \varepsilon \tag{1}$$

where k1 and k2 are the ontogenetic allometric coefficients; a, b, and c are allowed to vary randomly from child to child; and Log ($\varepsilon$), is assumed to have a constant error variance. The constant a is also allowed to vary for different populations, in this case the fixed factor: sex.

The model has the advantage of having proportional body size components and a multiplicative error that assumes they will increase proportionally with the physical performance variable Y (e.g., see Figure 1a,b).

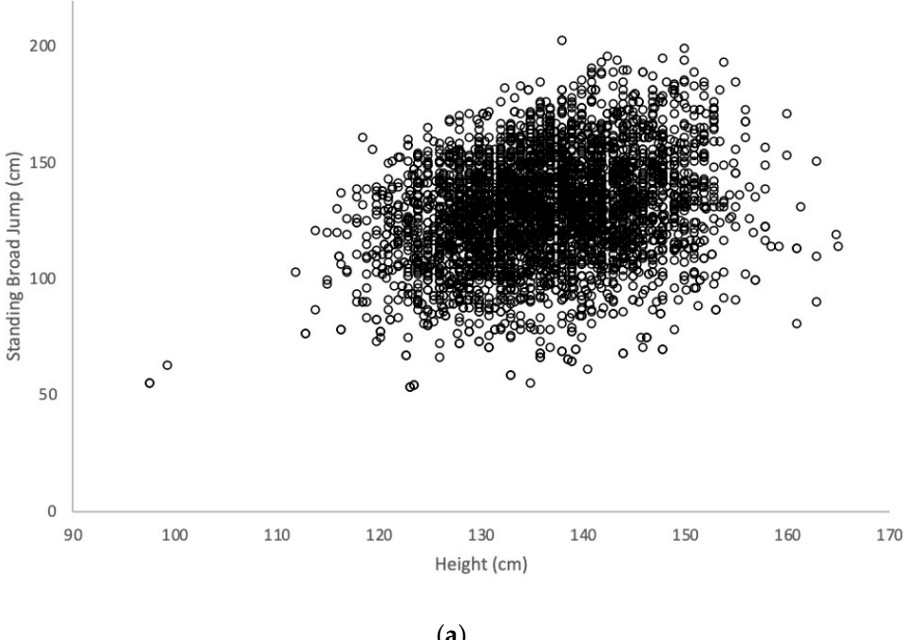

(**a**)

**Figure 1.** *Cont.*

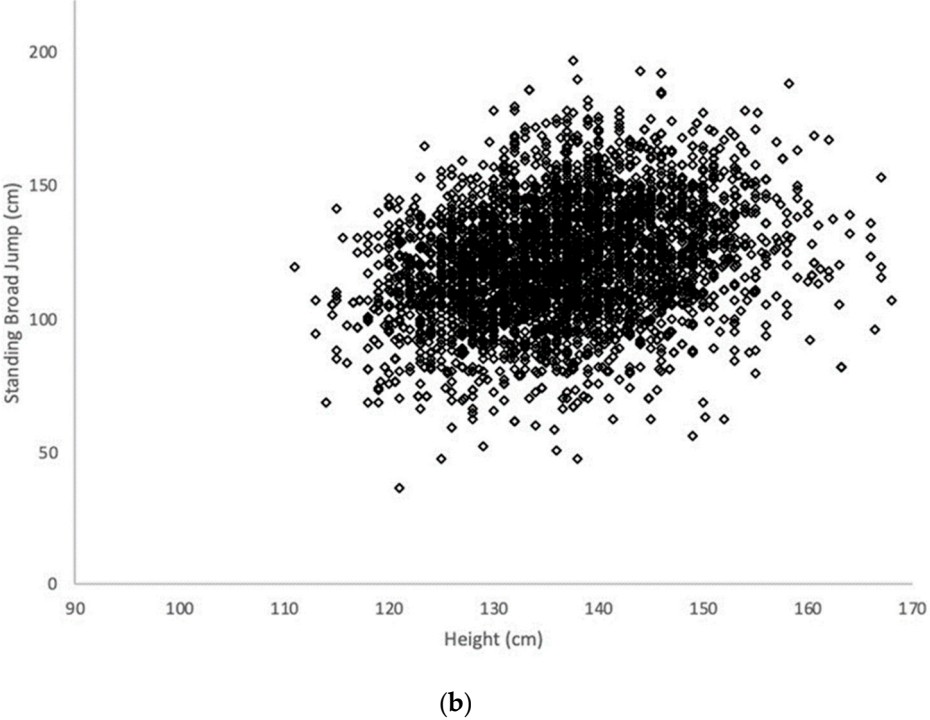

**(b)**

**Figure 1.** (**a**) The association between SBJ and height in 8–11-year-old male children, and (**b**) the association between SBJ and height in 8–11-year-old female children.

The model (Equation (1)) can be linearized with a log transformation (Equation (2)). A linear regression analysis or analysis of covariance (ANCOVA) on log(Y) can then be used to estimate the unknown parameters of the log-transformed model:

$$\text{Log(Y)} = \log(a) + k1 \cdot \log(M) + k2 \cdot \log(H) + c \cdot MO + \log(\varepsilon) \tag{2}$$

Further categorical differences within the population (e.g., sex and age) can be explored by allowing the constant intercept parameter log(a) to vary for each group by introducing them as fixed factors (plus possible interactions) within an ANCOVA. The significance level was set at $p < 0.05$.

## 3. Results

The explosive strength expressed by the SBJ revealed a mean value (SD) of 128.29 cm (22.76) and 121.17 cm (21.49), while the MO was, on average, 3.36 y (0.72) and 6.35 y (0.44) for boys and girls, respectively. A significant difference between sex was found ($p < 0.001$). Figure 2 shows the results between sexes (with the covariates factored in), suggesting that females outperformed males in the SBJ; females had greater performances than males by up to 20 cm (on average). Furthermore, the parameter Log a that results from interaction between height, mass, MO, sex, and age was 0.973, 0.985, 0.982, and 0.995 for males and 1.199, 1.211, 1.208, and 1.211 for females, respectively, for 8, 9, 10, and 11-year-olds.

The estimated parameters from the multiplicative model relating the SBJ distance to the body size components in Equation (1), incorporating the MO, are given in Table 2.

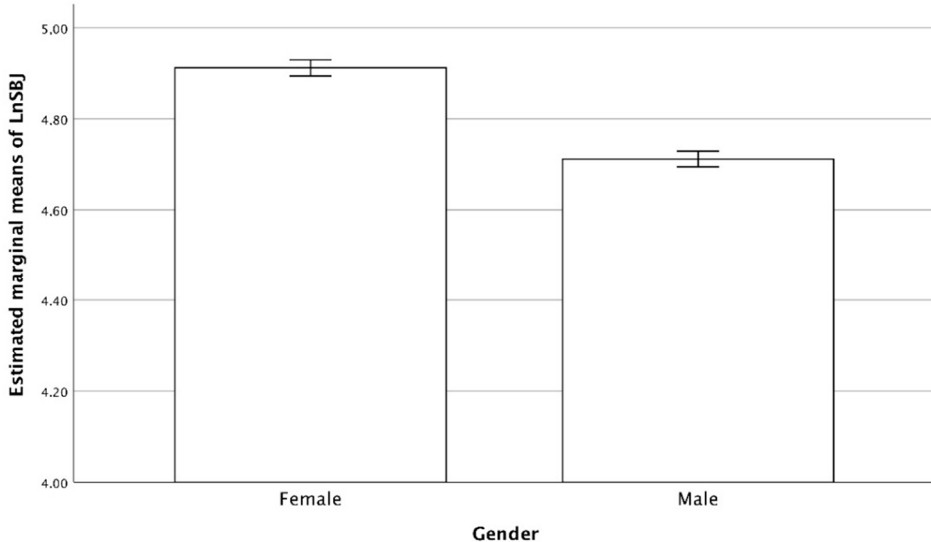

**Figure 2.** Estimated marginal means of SBJ by sex. Covariates appearing in the model are evaluated as follows: Ln Height = 4.9147, LN Mass = 3.4928, and MO = −4.8206.

**Table 2.** Parameters of Equation (1) about SBJ performance. The parameter Log a for males is 0.973, 0.985, 0.982, and 0.995, and 1.199, 1.211, 1.208, and 1.211 for females, respectively, for 8, 9, 10, and 11-year-olds. Note that the baseline group is the 11-year-old boys from which all other sex and age groups are compared.

| Parameter | B | Std. Error | t | Sig. | 95% Confidence Interval | |
|---|---|---|---|---|---|---|
| | | | | | Lower Bound | Upper Bound |
| Intercept Log(a) | 0.995 | 0.327 | 3.043 | 0.002 | 0.354 | 1636 |
| Log(M) (k1) | 1.152 | 0.064 | 18.092 | 0.000 | 1027 | 1276 |
| Log(H) ($k_2$) | −0.435 | 0.015 | −28.935 | 0.000 | −0.465 | −0.406 |
| MO (c) | 0.084 | 0.012 | 7.155 | 0.000 | 0.061 | 0.107 |
| Sex (Female) | 0.226 | 0.043 | 5.311 | 0.000 | 0.143 | 0.310 |
| Age (8.00 y) | −0.022 | 0.019 | −1.205 | 0.228 | −0.059 | 0.014 |
| Age (9.00 y) | −0.010 | 0.014 | −0.704 | 0.482 | −0.037 | 0.017 |
| Age (10.00 y) | −0.013 | 0.009 | −1.376 | 0.169 | −0.032 | 0.006 |
| Age (8.00) *Sex (Female) | −0.083 | 0.017 | −4746 | 0.000 | −0.117 | −0.049 |
| Age (9.00) *Sex (Female) | −0.065 | 0.014 | −4631 | 0.000 | −0.092 | −0.037 |
| Age (10.00) *Sex (Female) | −0.009 | 0.012 | −0.785 | 0.432 | −0.032 | 0.014 |

The model (Equation (1)) relating the SBJ distance to the body size components was

$$\text{SBJ distance (cm)} = a \cdot M^{-0.435} \cdot H^{1.152} \tag{3}$$

With a positive height (H) and negative mass (M), the model suggested that the optimal body shape associated with the SBJ is to be taller, but lighter (less body mass).

Fittingly, the model (Equation (1)) revealed significant differences in the constant Log a parameter (Table 2) due to sex ($p < 0.001$) and age ($p < 0.001$), together with the interactions of age and sex ($p < 0.001$, Figure 3).

In particular, the parameter Log a appears in the footnote of Table 2. Note that the maturity offset (MO) made significant positive contributions to predicting the log-transformed SBJ distance ($p < 0.001$).

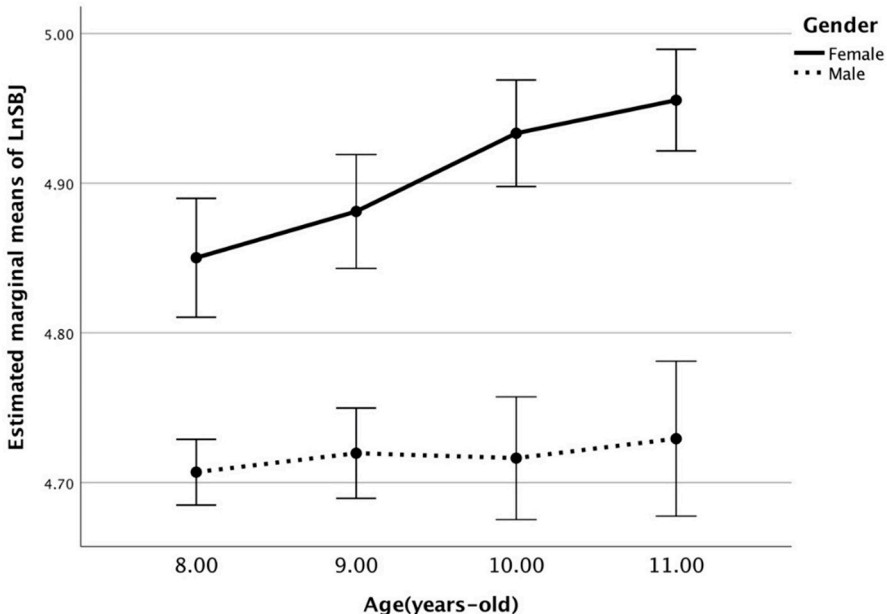

**Figure 3.** Estimated marginal means of the SBJ according to sex and age group. Covariates appearing in the model are evaluated as follows: LN height = 4.9147, LN weight = 3.4928, and MO = −4.8206.

## 4. Discussion

Since childhood, musculoskeletal fitness evaluation has been a crude indicator of both child performance and general health. To evaluate muscular power, the SBJ was included in all field test batteries. Researchers created reference values to characterize growth curves, not only in terms of anthropometrics (body mass and height) but also regarding physical performance. A close relation does exist, in fact, between the BMI and SBJ performance. In contrast, recent studies found that the BMI has limitations for measuring growth during childhood. From the age of seven to eight, body mass and height did not follow the same growth trends, making the BMI an improper predictor of SBJ performance [11,31]. In this light, this study aimed to investigate SBJ performance according to a robust scaling in which height and body mass are objectively valued, considering their actual trends owing to PHV assessment as a crucial individual factor of all physical performances. The data of this study confirmed the efficacy of the allometric approach to identify the most appropriate body size and shape characteristics associated with SBJ performance through the PHV. For boys and girls 8–11 years old, it is the ectomorph one (M = −0.435; H = 1.1152). In fact, in explosive strength, the MO contributed to predicting log-transformed SBJ distance since childhood. Without an allometric approach, consistent with other studies [13,14], child SBJ performance differs between boys and girls, with boys outperforming in a constant growth trend. Instead, the introduction of the size approach through allometric modeling to investigate SBJ performance with MO evaluation allowed us to refine the analysis, releasing different results. Males throughout the studied time frame (8–11 years) did not significantly augment their performances, whereas females showed a significant increase with a better SBJ performance (Figure 2). In line with other results [31,32], males did not outperform females in SBJ performance until reaching 11 years old. Furthermore, Newton's second law suggests that the force acting on a body is directly proportional to acceleration and shares its direction and orientation, with a proportionality constant given by the mass of the body. With that being said, Martin et al. [33] also showed that during maturation, female peak power is most likely to be determined by the quantitative properties of the muscle. Even in our study, the influence of body mass and related MO on strength performance in females was supported by the significant relationship that emerged between the SBJ and PHV offset. Moreover, Armstrong et al. [18] identified that, during the growth period, an index of lower limb power (assessed through the Wingate test) is strongly linked with both body mass and

fat-free mass. This is probably explained by a lower body mass of the participants, which allows them to perform more efficiently. However, the possibility of a genuine comparison is hard due to the lack of data investigated with allometric approaches in the same age group. A small number of studies investigated such a performance during the early stages of maturity. Dos Santos et al. [34] suggested a better SBJ performance in 10–15-year-old boys controlled by body and shape with respect to their female peers. On the other hand, Meylan et al. [21] suggested that, when maturity was adjusted, the horizontal jump length results were slightly better ($p = 0.04$) for males than females.

Overall, the results of this study highlighted that females outperformed males in SBJ performance. A putative explanation may reside in the lower mass and higher coordination level of the females, with respect to males. Although our study did not investigate coordination, D'Hondt et al. [31] found an inverse correlation between motor coordination and lower mass in children of 5–10 years, and Stodden et al. [35] reported that children with a higher level of motor coordination exhibit greater performance (explosive strength). In this sense, sex differences in lower limb power performance have been related to the expression of significant differences in fat-free mass [36]. Subsequently, changes in force production at the age of 13–14 years have been attributed to the dramatic increase in steroid concentrations in males [37]. From a body shape view, the explosive strength in the lower limbs is facilitated through an ectomorphic body shape [23]. This is of relevance since these values are similar to those found by Lovecchio et al. [11] in middle school students (11–14 years old, Log(M) = −0.357, Log(H) = 1.302). Interestingly, in the two studies the trend was comparable, as though the explosive force paralleled the growth factors. Furthermore, the values of the a parameter for males and females (Table 2) are very practical, as they effectively complete the predictive equation for a field test widely used to evaluate physical fitness and, possibly, related general health beyond the performance extent by itself. Future studies may concern other field tests very commonly employed in children, such as speed and agility tests.

## 5. Conclusions

This study suggested some practical applications for young people in the early stages of maturation. First, coaches of young athletes must be aware of the sex differences because of maturity issues; therefore, females with lower body masses than males may be favored in lower limb performance. Second, coaches should be adopting tests and sample-specific scaling modeled after body mass, rather than using theoretical models based on the assumption of body dimension similarity. From this perspective, the use of the parameter a (Table 2) would be of meaningful assistance. Third, sex-specific training may also be needed, given the early PHV in girls. This should make coaches and physical education teachers aware that females can train strength earlier than males. Furthermore, this approach was aligned with the bio-banding rationale that employed a stratification per maturity offset, within a competitive context (and not only assessment procedure or in-training evaluation) [38,39]. Finally, age, maturity offset, body shape, and body mass should be carefully considered when interpreting children's lower limb performance, as they are relevant indicators to the growth curves.

**Author Contributions:** Conceptualization, N.L., M.G., and A.M.N.; methodology, R.C., G.C., and N.L.; validation, M.G., M.V., and A.M.N.; investigation, R.C. and M.V.; data curation, M.G. and M.V.; writing—original draft preparation, R.C. and G.C.; writing—review and editing, N.L. and M.G.; supervision, N.L. and A.M.N.; project administration, V.C.P., R.C., and M.V. All authors have read and agreed to the published version of the manuscript.

**Funding:** This research received no external funding.

**Acknowledgments:** The authors would like to acknowledge all the students, teachers, and staff of the schools, university students, and graduates. All authors disclose professional relationships with companies or manufacturers who will benefit from the results of the present study.

**Conflicts of Interest:** The authors declare no conflict of interest.

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
