# Peer review of "Explosive Strength Modeling in Children: Trends According to Growth and Prediction Equation"

_applsci, doi:10.3390/app10186430_

Round 1
Reviewer 1 Report
Comments for the author
applsci-928252
Title: Explosive strength modeling in children: trends according to growth and prediction equation
Article Type: Article
Keywords: allometry, standing broad jump, children, growth, maturity offset
GENERAL:
This study tries to model strength test-specific allometric equations considering multiple individual factors such as height, body mass, sex, and age. Although many studies have been conducted using allometric approaches to correct field test performance's raw data, most of them are using one-variable models while multivariable models are in lack. Therefore, this study is a welcome addition to the literature. The study itself has been accomplished in a suitable and straightforward manner, and there is little to comment on about the introduction, the methods employed, or discussion of the findings. There are also some minor comments and suggestions, which are detailed below.
SPECIFIC:
Abstract
Line 17-18: It appears that “then the” may be unnecessary in this sentence. Consider removing it
Line 19: It seems that you are missing a comma “which”. Consider adding a comma “, which”.
Line 27: It appears that you type “hape”. Consider correct it to “shape”.Line 28: It seems that you are missing a comma “whereas”. Consider adding a comma “, whereas”. Introduction
Line 50: It seems that you are missing a comma “Tomkinson et al. [10]”. Consider adding a comma “Tomkinson et al. [10],”.
Line 51: It seems that you are missing a comma “while”. Consider adding a comma “, while”.
Line 51-52: Your sentence “ 10-11 years (both males and females) while… to age group and sex” may be hard to follow. Consider rephrasing to “10-11 years (both males and females), while no information was obtained about changes in body mass and height changes, which differ according to age group and sex”.
Line 53: It appears that “Kind of” may be unnecessary in this sentence. Consider removing it
Line 54: It seems that you are misusing the word “age”. Consider correct it to “period”.
Line 60: It seems that you are missing an article “misleading”. Consider adding an article “a misleading”.
Line 61: It seems that you are missing an article “to social”. Consider adding an article “to the social”.
Line 62: It seems that you have two independent clauses improperly joined with comma “categories [16], however,…”. Consider correcting the comma splice “categories [16]; however,…” or “categories [16]. However,…”.Line 62: It appears that the preposition “as” may be incorrect in this context. Consider changing it to “in”.
Line 65: It appears that “In fact” may be unnecessary in this sentence. Consider removing it.
Line 66-69: Your sentence “In this framework, it is thus … also considering the exponential trend of human growth [22, 23]” may be hard to follow. Consider rephrasing to “In this framework, it is crucial … factors, considering the exponential trend of human growth [22, 23]”.
Line 74: It seems that you are missing a comma “whereas”. Consider adding a comma “, whereas”.
Line 78: The phrase “taking in to account” may be wordy. Consider changing the wording to “considering”.
Line 80: It appears that the “a” in the “a reliable” is unnecessary in this context. Consider removing it.
Methods
Line 85: It seems that you are missing a comma “whereas”. Consider adding a comma “, whereas”.
Line 87: It seems that “children” may be incorrect. Consider changing the wording to “children’s”
Line 88: It seems that you are missing a comma “taking”. Consider adding a comma “, taking”.
Line 90: The phrase “by means of” may be wordy. Consider changing the wording to “using” or “utilizing” or “employing”.
Line 92: It seems that you have an unnecessary comma “"early", or ". Consider removing comma.
Line 96: It appears that “a total of” may be unnecessary in this sentence. Consider removing it.
Line 107-8: It appears that “that were” may be unnecessary in this sentence. Consider removing it.
Line 116 and 120: It seems that you are missing a comma “and”. Consider adding a comma “, and”.
Line 123: It seems that you have an unnecessary comma “classes,". Consider removing the comma.
Line 123: It appears that the singular verb “was” does not agree with the plural compound subject “Previous training and calibration of the operators”. Consider changing the wording to “were”.
Line 125: It appears that the singular verb “was” does not agree with the plural compound subject “The presence and collaboration of the curricular teachers”. Consider changing the wording to “were”.
Results
Line 173: It seems that semicolon use may be incorrect here “(21.49);”. Consider using a comma instead of a semicolon.
Discussion
Line 206: The comma before the conjunction “but also” appears to be unnecessary. Consider correct it.
Line 204-206: Your sentence “ To evaluate muscular power… but also as regards physical performance.” may be hard to follow. Consider rephrasing to “To evaluate muscular power, SBJ was included in all field test batteries. Researchers created reference values to characterize growth curves, not only in terms of anthropometrics (body mass, height) but also as regards physical performance”.
Line 218: It seems that you are missing a comma “with”. Consider adding a comma “, with”.
Line 219: It seems that you have an unnecessary comma “evaluation,". Consider removing the comma.
Line 221: It seems that you have an unnecessary comma “,did". Consider removing the comma.
Line 221: It seems that you are missing a comma “whereas”. Consider adding a comma “, whereas”.
Line 232: It seems that you are missing a comma “which”. Consider adding a comma “, which”.
Line 232: It seems that the verb “allow” does not agree with the subject. Consider changing the verb form to “allows”.
Line 244: It appears that “likewise a” may be unnecessary in this sentence. Consider removing it.
Line 245: It appears that the preposition “with” may be incorrect in this context. Consider changing it to “to”.
Conclusions
Line 259: This sentence appears to consist of two independent clauses. Consider using a semicolon or period before “therefore”.
Line 266: It appears that an article is missing before the word “competitive”. Consider adding the article “a”.
Line 266: It appears that the preposition “in” is redundant. Consider removing it.
Additionally, I couldn't find the table 1 as mentioned in the text, so I didn't review it.
Author Response
To Editor–in-Chief of Applied Sciences,
Special Issue: "New Trends in Sport and Exercise Medicine”
Revision: Manuscript applsci-928252 for special issue “New Trends in Sport and Exercise Medicine”: titled “Explosive strength modeling in children: trends according to growth and prediction equation” by Carnevale Pellino V et.al.
Dear Editor in Chief,
We were encouraged by the fact that reviewers found the paper of interest. For this reason, the paper has now been modified, and revised in accordance with the reviewer’s comments (major changes in red font).
We have addressed the points raised reviewers in detail below.
During the revision we discover a major mistake in our submission. Unfortunately, we forgot in authorship an author. This is an error imputable to us and we upgrade the list with a new last author. We are very sorry for the inconvenience.
The authors declare no conflict of interest.
We earnestly hope that the revised manuscript will now be suitable for publication in Applied Sciences.
Sincerely yours
Dr. Gabriele Ceccarelli
Human Anatomy Institute, Department of Public Health, Experimental Medicine & Forensic Science, University of Pavia, Pavia, Italy
Response to Reviewer 1 Comments
This study tries to model strength test-specific allometric equations considering multiple individual factors such as height, body mass, sex, and age. Although many studies have been conducted using allometric approaches to correct field test performance's raw data, most of them are using one-variable models while multivariable models are in lack. Therefore, this study is a welcome addition to the literature. The study itself has been accomplished in a suitable and straightforward manner, and there is little to comment on about the introduction, the methods employed, or discussion of the findings. There are also some minor comments and suggestions, which are detailed below.
We thank the reviewer for the comment; it is always very rewarding to read these appreciations. We will correct all the minor comments detailed below.
Abstract
Line 17-18: It appears that “then the” may be unnecessary in this sentence. Consider removing it
Done
Line 19: It seems that you are missing a comma “which”. Consider adding a comma “, which”.
Done
Line 27: It appears that you type “hape”. Consider correct it to “shape”.Line 28: It seems that you are missing a comma “whereas”. Consider adding a comma “, whereas”.
Done
Introduction
Line 50: It seems that you are missing a comma “Tomkinson et al. [10]”. Consider adding a comma “Tomkinson et al. [10],”.
Done
Line 51: It seems that you are missing a comma “while”. Consider adding a comma “, while”.
Done
Line 51-52: Your sentence “ 10-11 years (both males and females) while… to age group and sex” may be hard to follow. Consider rephrasing to “10-11 years (both males and females), while no information was obtained about changes in body mass and height changes, which differ according to age group and sex”.
Done
Line 53: It appears that “Kind of” may be unnecessary in this sentence. Consider removing it
Done
Line 54: It seems that you are misusing the word “age”. Consider correct it to “period”.
Done
Line 60: It seems that you are missing an article “misleading”. Consider adding an article “a misleading”.
Done
Line 61: It seems that you are missing an article “to social”. Consider adding an article “to the social”.
Done
Line 62: It seems that you have two independent clauses improperly joined with comma “categories [16], however,…”. Consider correcting the comma splice “categories [16]; however,…” or “categories [16]. However,…”.Line 62: It appears that the preposition “as” may be incorrect in this context. Consider changing it to “in”.
Done
Line 65: It appears that “In fact” may be unnecessary in this sentence. Consider removing it.
Done
Line 66-69: Your sentence “In this framework, it is thus … also considering the exponential trend of human growth [22, 23]” may be hard to follow. Consider rephrasing to “In this framework, it is crucial … factors, considering the exponential trend of human growth [22, 23]”.
Done
Line 74: It seems that you are missing a comma “whereas”. Consider adding a comma “, whereas”.
Done
Line 78: The phrase “taking in to account” may be wordy. Consider changing the wording to “considering”.
Done
Line 80: It appears that the “a” in the “a reliable” is unnecessary in this context. Consider removing it.
Done
Methods
Line 85: It seems that you are missing a comma “whereas”. Consider adding a comma “, whereas”.
Done
Line 87: It seems that “children” may be incorrect. Consider changing the wording to “children’s”
Done
Line 88: It seems that you are missing a comma “taking”. Consider adding a comma “, taking”.
Done
Line 90: The phrase “by means of” may be wordy. Consider changing the wording to “using” or “utilizing” or “employing”.
Done
Line 92: It seems that you have an unnecessary comma “"early", or ". Consider removing comma.
Done
Line 96: It appears that “a total of” may be unnecessary in this sentence. Consider removing it.
Done
Line 107-8: It appears that “that were” may be unnecessary in this sentence. Consider removing it.
Done
Line 116 and 120: It seems that you are missing a comma “and”. Consider adding a comma “, and”.
Done
Line 123: It seems that you have an unnecessary comma “classes,". Consider removing the comma.
Done
Line 123: It appears that the singular verb “was” does not agree with the plural compound subject “Previous training and calibration of the operators”. Consider changing the wording to “were”.
Done
Line 125: It appears that the singular verb “was” does not agree with the plural compound subject “The presence and collaboration of the curricular teachers”. Consider changing the wording to “were”.
Done
Results
Line 173: It seems that semicolon use may be incorrect here “(21.49);”. Consider using a comma instead of a semicolon.
Done
Discussion
Line 206: The comma before the conjunction “but also” appears to be unnecessary. Consider correct it.
Done
Line 204-206: Your sentence “ To evaluate muscular power… but also as regards physical performance.” may be hard to follow. Consider rephrasing to “To evaluate muscular power, SBJ was included in all field test batteries. Researchers created reference values to characterize growth curves, not only in terms of anthropometrics (body mass, height) but also as regards physical performance”.
Done
Line 218: It seems that you are missing a comma “with”. Consider adding a comma “, with”.
Done
Line 219: It seems that you have an unnecessary comma “evaluation,". Consider removing the comma.
Done
Line 221: It seems that you have an unnecessary comma “,did". Consider removing the comma.
Done
Line 221: It seems that you are missing a comma “whereas”. Consider adding a comma “, whereas”.
Done
Line 232: It seems that you are missing a comma “which”. Consider adding a comma “, which”.
Done
Line 232: It seems that the verb “allow” does not agree with the subject. Consider changing the verb form to “allows”.
Done
Line 244: It appears that “likewise a” may be unnecessary in this sentence. Consider removing it.
Done
Line 245: It appears that the preposition “with” may be incorrect in this context. Consider changing it to “to”.
Done
Conclusions
Line 259: This sentence appears to consist of two independent clauses. Consider using a semicolon or period before “therefore”.
Done
Line 266: It appears that an article is missing before the word “competitive”. Consider adding the article “a”.
Done
Line 266: It appears that the preposition “in” is redundant. Consider removing it.
Done
Additionally, I couldn't find the table 1 as mentioned in the text, so I didn't review it.
Sorry for the failure to upload Table 1. We will add Table 1 in page 3, lines 127-128.

Reviewer 2 Report
I was becoming confused by the way the stats were described in the methods and the tables (and I am a statistisical reviewer). Keep the maths lessons out of it and just make it clear what was done and why and how the results were summarised. EG Univariable and multivariable linear regression was used to examine the the effect of ht, wt, MI, sex on SBJ (you should do the univariable models for each covariate to be able to determine individual effects as well - including R2 ). What interactions eg age*sex were added into the models.
What was the original distribution of the data? Was log-transforming the most appropriate thing to do- how was this checked? Were model residuals checked for normality- what was done to check model fit?
I suggest just present the B(seB) (95%CI), R2 and marginal means of the outcome for each predictor in the table.
Author Response
To Editor–in-Chief of Applied Sciences,
Special Issue: "New Trends in Sport and Exercise Medicine”
Revision: Manuscript applsci-928252 for special issue “New Trends in Sport and Exercise Medicine”: titled “Explosive strength modeling in children: trends according to growth and prediction equation” by Carnevale Pellino V et.al.
Dear Editor in Chief,
We were encouraged by the fact that reviewers found the paper of interest. For this reason, the paper has now been modified, and revised in accordance with the reviewer’s comments (major changes in red font).
We have addressed the points raised reviewers in detail below.
During the revision we discover a major mistake in our submission. Unfortunately, we forgot in authorship an author. This is an error imputable to us and we upgrade the list with a new last author. We are very sorry for the inconvenience.
The authors declare no conflict of interest.
We earnestly hope that the revised manuscript will now be suitable for publication in Applied Sciences.
Sincerely yours
Dr. Gabriele Ceccarelli
Human Anatomy Institute, Department of Public Health, Experimental Medicine & Forensic Science, University of Pavia, Pavia, Italy
Response to Reviewer 2 Comments
I was becoming confused by the way the stats were described in the methods and the tables (and I am a statistisical reviewer). Keep the maths lessons out of it and just make it clear what was done and why and how the results were summarised. EG Univariable and multivariable linear regression was used to examine the the effect of ht, wt, MI, sex on SBJ (you should do the univariable models for each covariate to be able to determine individual effects as well - including R2 ). What interactions eg age*sex were added into the models.
Dear Reviewer, thank you for your comment, I think they can make the paper clearer and more understandable. We add modified Table 2 to incorporate the age by sex interaction parameters. These results can also be clearly seen in Figure 3
What was the original distribution of the data? Was log-transforming the most appropriate thing to do- how was this checked? Were model residuals checked for normality- what was done to check model fit?
- With such a large sample of 7,823 children, all test of normality were significant whether we log-transformed or not. However, given that the majority of data are all ratio measurements (SBJ, H, M cannot be negative), the multiplicative nature of model (Eq. 1) ensures that the prediction equation (Eq. 1) can never predict negative SBJ score. This would not be the case if we had fitted an additive rather than a multiplicative model.
